

**MONITORING POTENTIAL IONOSPHERE CHANGES CAUSED BY**
**VAN EARTHQUAKE (Mw 7.2) USING GNSS MEASUREMENTS**
**Selcuk PEKER[1], Samed INYURT [2]and Cetin MEKIK[2]**
[1]General Command of Mapping, Ankara, Turkey
[2]Bulent Ecevit University, Geomatics Engineering Department, Zonguldak
**ABSTRACT**
Several scientists from different disciplines have studied earthquakes for many years. As a
result of these studies, it has been proposed that some changes take place in the ionosphere layer
before, during or after earthquakes, and the ionosphere should be monitored in earthquake
prediction studies. This study investigates the changes in the ionosphere created by the
earthquake with magnitude of Mw=7.2 in the northwest of the Lake Erçek which is located to
the north of the province of Van in Turkey on 23 October 2011 and at 1.41 pm local time (-3
UT) with the epicenter of 38.758° N, 43.360° E using the TEC values obtained by the Global
Ionosphere Models (GIM) created by IONOLAB-TEC and CODE. In order to see whether the
ionospheric changes obtained by the study in question were caused by the earthquake or not,
the ionospheric conditions were studied by utilizing indices providing information on solar and
geomagnetic activities (F10.7 cm, Kp, Dst).
As a result of the statistical test on the TEC values obtained from the both models, positive and
negative anomalies were obtained for the times before, on the day of and after the earthquake,
and the reasons for these anomalies are discussed in detail in the last section of the study. As
the ionospheric conditions in the analyzed days were highly vibrant, it was thought that the
anomalies were caused by geomagnetic effects, solar activity and the earthquake. The authors
believe that interdisciplinary studies are needed to distinguish the earthquake-related part of the
anomalies in question.
**Keywords:** TEC, Van Earthquake, Ionosphere












## 1. INTRODUCTION


The ionosphere is the part of the atmosphere at the altitudes of 60 km to 1.100 km where there
are ions and free electrons in considerable amounts that can reflect electromagnetic waves
(URL-1). It completely covers the thermosphere, one of the main layers of the atmosphere, but
also includes some of the mesosphere and the exosphere.

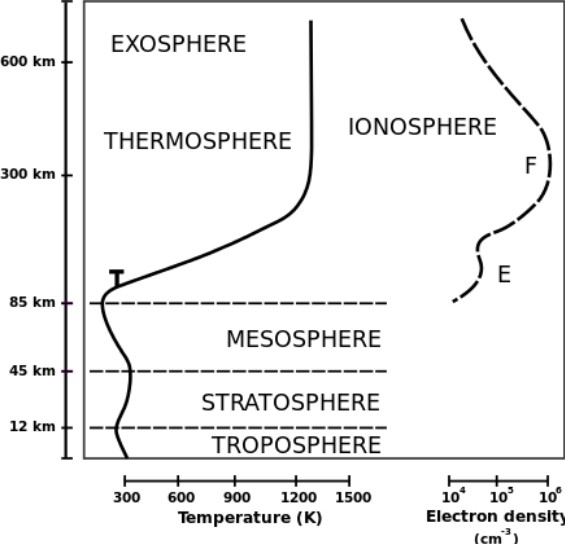


**Figure 1.** The Relationship between the Atmosphere and the Ionosphere (URL-2)

The atmospheric gas molecules in the layer in question are charged with electricity influenced
by the UV radiation from sunlight and disintegrating into ions and electrons, hence leading to
ionization. As a result of this ionization, especially the fields of information and communication
technologies, intensely using audio, data and signal interactions, are affected by the ionosphere
(Anderson and Fuller-Rowell, 1999)

When a radio wave reaches the ionosphere, the electric component of the electromagnetic wave
forces the electrons in the ionosphere to vibrate in the same frequency as the radio wave. The
vibration energy leads to reorganization of the electrons or the electrons' replication of the
original radio frequency. If the collision frequency of the ionosphere is lower than the radio
frequency and the quantity of electrons is sufficient, complete reflection occurs. If the frequency





of the radio wave is higher than the plasma frequency of the ionosphere, electrons cannot
provide feedback fast enough, and thus the signal is not reflected.

The most important parameter that defines the ionosphere in space and time is the quantity of
electrons. This quantity varies under the influence of the day-night cycle, seasons, geographical
location and magnetic storms in the sun. As it is not possible to measure the quantity of electrons
in the ionosphere directly, indirect measurement and calculation methods have been developed.
Total Electron Content (TEC), which is defined as the quantity of free electrons along a cylinder
with a cross section of 1 $m^2$, is a suitable parameter to monitor the changes in the ionosphere in
space and time. All signals that contain audio and data that pass through or get reflected from
the ionosphere, which is highly irregular and difficult to model, are affected by the structure of
this layer.

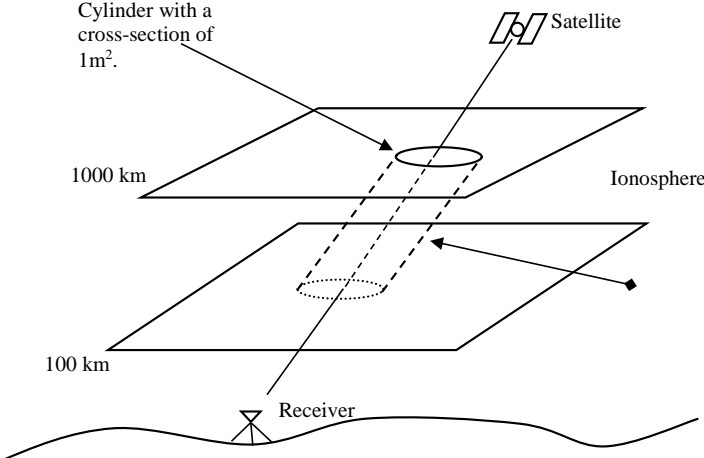


**Figure 2.** Graphical Representation of Total Electron Content (Langley, 2002)

Calculating Total Electron Content (TEC) is a method used directly to investigate the structure
of the ionosphere. TEC is represented by the unit of TECU, and one TECU equals to
$10^{16} \ el/m^2$ (Schaer, 1999). TEC is expressed in two ways: STEC (Slant Total Electron
Content); the free electron content calculated along the slanted line between the receiver and
the satellite, and VTEC (Vertical Total Electron Content); the free electron content calculated
along the zenith of the receiver (Langley, 2002).





TEC varies based on positional and temporal variables such as the latitude of the place, seasons,
solar activity, geomagnetic storms and earthquakes. Ionospheric altitude also differs based on
geography.
TEC, which is defined as the number of free electrons on the one square meter area on the line
followed by a radio wave, is one of the important parameters for examining the structure of the
ionosphere and the upper atmosphere. With TEC values, it is possible to examine the short and
long-term changes in the ionosphere, ionospheric irregularities and disruptive factors together
(Erol and Arıkan 2005; Başpınar 2012).
The changes in the ionosphere created by earthquakes were first studied in early 1960s. In order
to detect any prior sign before earthquakes, experts examined the critical frequency, the
maximum electron density in the F2 layer and total electron content (Yildirim et al., 2016).
Some studies have shown that ionospheric anomalies may be detected in a short time before
earthquakes using satellites (Pulinets 1998).
Ionospheric changes are being studies in more than twenty countries today as precursors of
earthquakes. Definition of ionospheric anomalies and feasibility studies of sismo-ionospheric
precursors are still ongoing (Yildirim et al., 2016).

**2. METHODOLOGY**

**2.1 IONOLAB-TEC Method:**

The IONOLAB-TEC method developed by the department of Electrical and Electronics
Engineering of Hacettepe University is a JAVA application that uses the Regularized TEC (D-
TEI) algorithm (Arikan et al. 2004 ).
In this application, they developed a method that estimates VTEC values by using all GPS
signals measured at a period of time in a day. While the measurements taken from the satellites
with elevations of $60^o$ or higher are used, the measurements from the satellites with elevations
of $10^o\ to\ 60^o$ are weighted by a Gauss function. The data from satellites with elevations of
lower than $10^o$ are not included in calculations to reduce multipath effects (Equations 1).

$$\omega_m(n) \begin{cases} 1, & 60^o \leq \in_m(n) \leq 90^o \\ \exp(-(60-\in_m(n))^2/2\sigma^2), & 10^o \leq \in_m(n) < 60^o \qquad (1) \\ 0, & \in_m(n) < 10^o \end{cases}$$

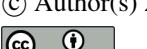




In the next step, the VTEC data obtained from satellites are combined by the least squares
method. For this, a cost function that will minimize the square of the error between the VTEC
data that will be obtained as a result of estimation, and the VTEC data calculated from the
satellites is defined as below.
$J_{\mu.k_c}(x) = \sum_{m=1}^{M}(x - x_m)^2 W_m(x - x_m) + \mu x^T H(k_c)$    (2)
$W_m = diag(w_m),$
$H(k_c)$ = Filter that allows components of frequencies up to $k_c$ to pass,
$\mu$ = Regularization coefficient.
To find the $x$ estimates that minimize the error, if the derivative of the statement in Equation
(2) is taken and equated to 0;
$(\Delta_x J_{\mu,k_c}(x)) = 0$    (3)
The minimization process for the cost function turns into the solution of a linear system like:
$A(\mu, k_c)x = b$    (4)
In the equation above:
$A(\mu, k_c) = \sum_{m=1}^{M} W_m + \mu H(k_c)$    (5)
$b = \sum_{m=1}^{M} W_m x_m$    (6)
Therefore, the VTEC estimates $\widetilde{\mathbf{X}}$ are found as,
$\tilde{X}(\mu, k_c) = A^{-1}(\mu, k_c)b$    (7)
The high-pass penalty filter used for TEC estimates may be organized as the $\mathbf{H}(k_c)$ Toeplitz
matrix:
$\mathbf{H}(k_c) = \begin{bmatrix} h_0(k_c) & h_1(k_c) \Lambda & h_{N-1}(k_c) \\ h_{N-1}(k_c) & h_0(k_c) \Lambda & h_{N-2}(k_c) \\ M & M & M \\ h_1(k_c) & h_2(k_c) \Lambda & h_0(k_c) \end{bmatrix}$    (8)
$h_n(k_c) = \frac{1}{N}\sum_{k=0}^{N-1} H_k(\omega_c)\exp(j\frac{2\pi}{N}kn)$    (9)
$\omega_c = 2\pi k_c/N$. The filter function $\mathbf{H}_k(\omega_c)$ may be chosen as in Equation (11).
$H_k(\omega_c) = \begin{cases} 1, & \text{if } \pi - \omega_c \leq \dfrac{2\pi}{N}k \leq \pi + \omega_c \\ 0, & \text{diger} \end{cases}$    (10)



$$h_n(k_c) = \begin{cases} 1 - \dfrac{1}{N}(2k_c+1), & n=0 \\ -sin\left(\dfrac{\pi n}{N}(2k_c+1)\right) \Big/ \left(N\,sin(\dfrac{\pi n}{N})\right), & n \neq 0 \end{cases}$$
(11)


The error function between the VTEC values calculated from satellites $x_m$ and the VTEC
estimates $\widetilde{x}$ is given in Equation (12). The operation ||.|| describes the norm statement of the
difference vector weighted between the VTEC estimates and calculations.
$$e(\mu, k_c) = \sum_{m=1}^{M} \left\| W_m (\widetilde{x} - x_m) \right\|^2$$
(12)


In order to regularize the estimate values even more, floating median filter may be used. The
length of the median filter is another parameter to be determined. With the estimated VTEC
values, post-estimation median filter was applied, and the error function between the VTEC
values is given in Equation (13).
$$e_f(N_f) = \left\| \widetilde{x} - \widetilde{x}_{N_f} \right\|^2$$
(13)


In Equation (13), $\widetilde{x}_{N_f}$ shows the $\widetilde{x}$ estimates processed with a median filter with the length of
$N_f$. For the method to work accurately, suitable $\mu$, $k_c$ and $N_f$ parameters must be determined. The
details provided up to now cover the regularization method for a period of 24 hours.

When there is an estimation of TEC for a limited period of time, the cost function is redefined
as in Equation (14).

$$J_{\mu,k_c}(x) = \sum_{m=1}^{M} (x-x_m)^T W_m (x-x_m) + \mu (x-at)^T H(k_c)(x-at)$$
(14)


In the equation, $a$ is the slope of the line and **t** is the time vector for the period of time. In order
to find **x** estimates that minimize the cost function, the derivative of this function is taken, and
the result is equated to zero. In this case, minimization of the cost function is turned into the
solution of a system of equations as in Equation (15).
$$A(\mu, k_c)\begin{bmatrix} x \\ a \end{bmatrix} = b$$
(15)

The matrix **A** in Equation (16) and the vector **b** in Equation (17) are calculated as,





$$\mathbf{A}(\mu,k_c)=\begin{bmatrix} \sum_{m=1}^{M}\mathbf{W}_m+\mu\mathbf{H}(k_c) & -\mu\mathbf{H}(k_c) \\ \mathbf{t}^T\mathbf{H}(k_c) & -\mathbf{t}^T\mathbf{H}(k_c)\mathbf{t} \end{bmatrix} \qquad (16)$$

$$\mathbf{b}=\begin{bmatrix} \sum_{m=1}^{M}\mathbf{W}_m\mathbf{x}_m \\ 0 \end{bmatrix} \qquad (17)$$

Using the equations above, the $\tilde{\mathbf{x}}$ values showing the **x** estimates are calculated as in Equation

177 (18).


$$\begin{bmatrix} \tilde{\mathbf{x}}(\mu,k_c) \\ a \end{bmatrix}=A^{-1}(\mu,k_c)b \qquad (18)$$

As a result, the proposed regularization method may be applied for both day-long and limited
periods of time (Arıkan et al. 2004).

**2.2 Global Ionosphere Model (GIM):**

Global Ionospheric Maps are published in the IONEX (IONosphere map EXchange) format in
a way that covers the entire world. The institutions that produce these maps in the world include
CODE (Center for Orbit Determination in Europe, Switzerland), DLR (Fernerkundungstation
Neustrelitz, Germany), ESOC (European Space Operations Centre, Germany), JPL (Jet
Propulsion Laboratory, California), NOAA (National Oceanic and Atmospheric
Administration, United States), NRCan (National Resources, Canada), ROB (Royal
Observatory of Belgium, Belgium), UNB (University of New Brunswick, Canada), UPC
(Polytechnic University of Catalonia, Spain), WUT (Warsaw University of Technology,
Poland) (Aysezen, 2008). In this study we used the GIM-TEC values produced by CODE in the
IONEX format. In the dates they were analyzed, the temporal resolution of the TEC values was
2 hours, while their positional resolution was 2.5º by latitude and 5º by longitude. In order to
calculate TEC values for a point whose latitude and longitude is known on the GIM-TEC maps
created by CODE using more than 300 GNSS receivers around the world, the 4 TEC values
that cover the point and the two-variable interpolation formula are given below (Schaer et al.

1998).

$E_{int}(\lambda_0 + p\Delta\lambda, \beta_0 + q\Delta\beta) = (1-p)(1-q)E_{0.0} + p(1-q)E_{1.0} + q(1-p)E_{0.1} + pqE_{1.1}$ \qquad (19)





p and q: $0 \le p, q < 1$.
$\Delta\lambda$ and $\Delta\beta$: Longitude and Latitude differences grid widths,
$\lambda_0 \ and \ \beta_0$: Initial longitude and latitude values,
$E_{0.0}, E_{1.0}, E_{0.1} \ ve \ E_{1.1}$ : TEC values known in neighboring points,
$E_{int}$: TEC value to be found.

**3. ANALYSIS TO DETERMINE EARTHQUAKE-RELATED TEC CHANGES**

In order to investigate earthquake-related TEC changes, the TEC values for the stations close
to the epicenters, HAKK, MALZ, OZAL and TVAN were procured using the IONOLAB-TEC
and GIM-TEC models. The correlation coefficient was obtained for the TEC values from both
models between the dates 13.10.2011 and 02.11.2011 for the stations above.

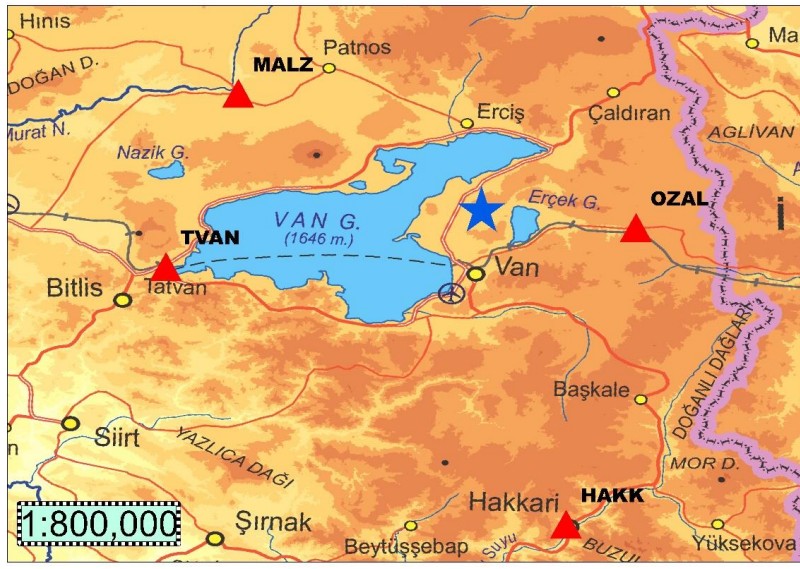


**Figure 3.** Analyzed Stations
Figure 3 shows the stations analyzed (represented by red triangles) and the epicenter of the
earthquake represented by blue star. For each station, the TEC values with the temporal
resolution of two hours obtained from both the IONOLAB-TEC and GIM-TEC models and the
correlation coefficient showing whether there is a linear relationship between two values were
calculated as below;

$r = \frac{\sum(xy) - (\sum x)(\sum y)/n}{\sqrt{(\sum x^2 - (\sum x)^2/n)(\sum y^2 - (\sum y)^2/n)}}$ (20)

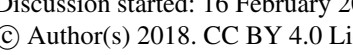



In order to determine the outlier values among the TEC values with a two-hour temporal
resolution from both models, the TEC values obtained from both models between the dates
01.10.2011 and 10.10.2011, which were considered calm in terms of geomagnetic and solar
activity, were used to determine the upper boundary (UB) and the lower boundary (LB). By
utilizing the TEC values from both models, the UB and LB values were calculated using the
formulae x+3σ and x-3σ. Here, x is the mean TEC value for the relevant epoch and σ is the
standard deviation. If the TEC value in any epoch is higher than the upper boundary, it is a
positive anomaly. Similarly if it is lower than the lower boundary, it is a negative anomaly. In
order to investigate whether the anomalies before, on the day of and after the earthquake were
caused by the earthquake or not, we also examined the Kp, Dst and F10.7 cm indices, which
provided information on the geomagnetic and solar activity for the days in which anomalies
were detected.

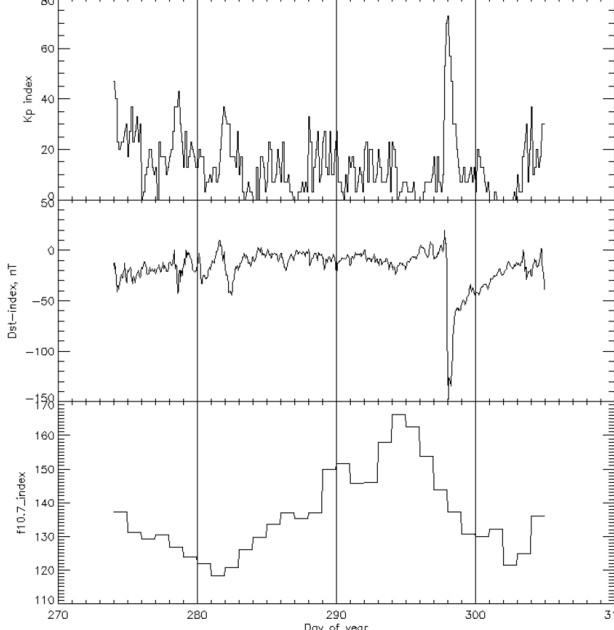



**Figure 4.** The Chart for October 2011 K*p,* Dst and F10.7 cm Indices (URL-3)

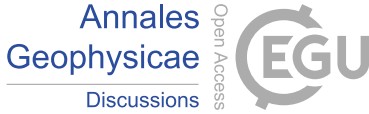

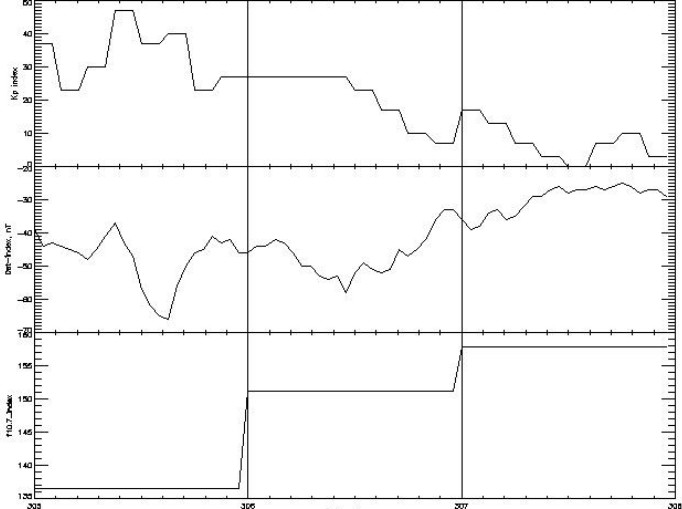


**Figure 5.** The Chart for the Dates 01-03.11.2011 in K*p,* Dst and F10.7 cm Indices (URL-3)

Figures 4 and 5 show the Kp, Dst and F10.7 cm indices that provide information on
geomagnetic and solar activity in October and on the first three days of November. Below are
the TEC values for the HAKK station for the dates 13.10.2011-02.11.2011 obtained using the
GIM-TEC and IONOLAB-TEC methods (Figures 6 and 7).

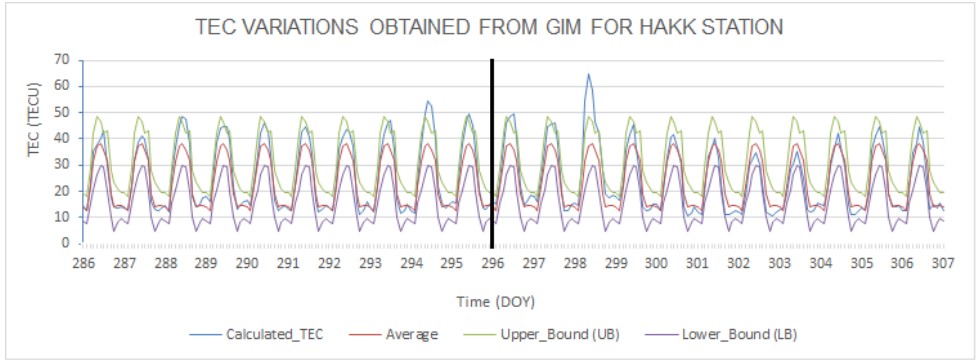


**Figure 6.** GIM-TEC Values for the HAKK Station

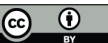





**Figure 7.** IONOLAB-TEC Values for the HAKK Station

The correlation coefficient *r* between the TEC values calculated by both methods for the HAKK station was 0.978469 indicating a strong positive relationship. The anomaly tables for this station are provided below (Tables 1 and 2).

| GIM-TEC Anomaly Table for HAKK Station | | | | | | | | | |
|---|---|---|---|---|---|---|---|---|---|
| Number | DOY | Hour | TEC Difference (TECU) | Type of Anomaly | Number | DOY | Hour | TEC Difference (TECU) | Type of Anomaly |
| 1 | 286 | 12 | 1.0 | Positive | 7 | 294 | 12 | 10.5 | Positive |
| 2 | 288 | 12 | 5.7 | Positive | 8 | 295 | 12 | 7.3 | Positive |
| 3 | 289 | 12 | 2.5 | Positive | 9 | 296 | 12 | 7.5 | Positive |
| 4 | 290 | 12 | 0.5 | Positive | 10 | 297 | 12 | 4.1 | Positive |
| 5 | 292 | 12 | 0.8 | Positive | 11 | 298 | 8 | 16.5 | Positive |
| 6 | 293 | 12 | 5.2 | Positive | | | | | |

**Table 1.** HAKK Station Global Ionosphere Model Anomaly Table

| IONOLAB-TEC Anomaly Table for HAKK Station | | | | | | | | | |
|---|---|---|---|---|---|---|---|---|---|
| Number | DOY | Hour | TEC Difference (TECU) | Type of Anomaly | Number | DOY | Hour | TEC Difference (TECU) | Type of Anomaly |
| 1 | 287 | 12 | 0.4 | Positive | 9 | 295 | 12 | 7.2 | Positive |
| 2 | 288 | 12 | 9.2 | Positive | 10 | 296 | 12 | 8.8 | Positive |
| 3 | 289 | 12 | 4.3 | Positive | 11 | 297 | 12 | 4.6 | Positive |
| 4 | 290 | 12 | 3.8 | Positive | 12 | 298 | 8 | 16.5 | Positive |
| 5 | 291 | 12 | 4.5 | Positive | 13 | 301 | 12 | 0.3 | Negative |
| 6 | 292 | 12 | 1.4 | Positive | 14 | 302 | 14 | 0.9 | Negative |
| 7 | 293 | 12 | 4.2 | Positive | 15 | 303 | 12 | 0.7 | Negative |
| 8 | 294 | 12 | 10.9 | Positive | 16 | 306 | 10 | 0.9 | Positive |

**Table 2.** HAKK Station IONOLAB-TEC Anomaly Table



Below are the TEC values for the MALZ station obtained using the GIM-TEC and IONOLAB-
TEC methods (Figures 8 and 9).

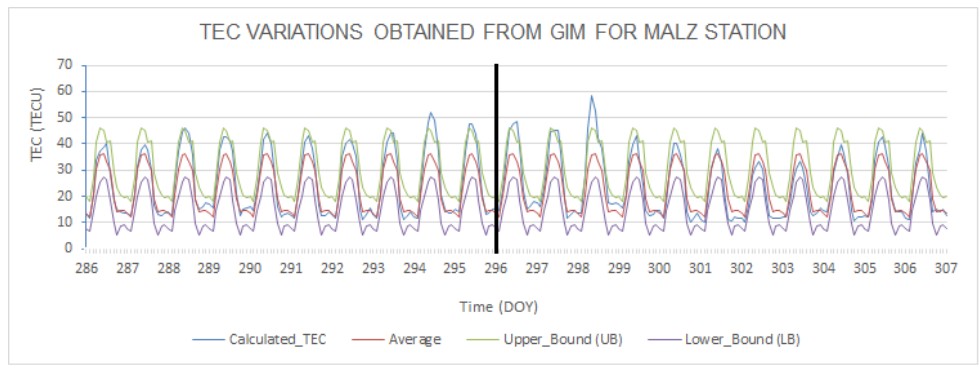

**Figure 8.** GIM-TEC Values for the MALZ Station

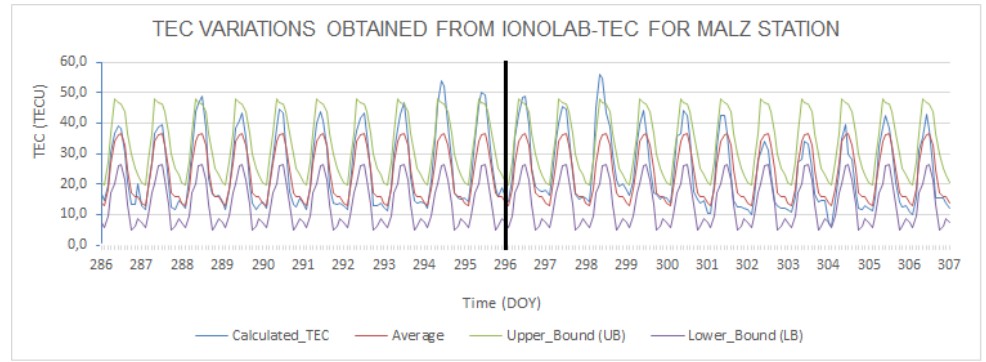

**Figure 9.** IONOLAB-TEC Values for the MALZ Station


The correlation coefficient *r* between the TEC values calculated by both methods for the MALZ
station was 0.976587 indicating also a strong positive relationship. The anomaly tables for this
station are provided below (Tables 3 and 4).

| GIM-TEC Anomaly Table for MALZ Station | | | | | | | | | |
|---|---|---|---|---|---|---|---|---|---|
| Number | DOY | Hour | TEC Difference (TECU) | Type of Anomaly | Number | DOY | Hour | TEC Difference (TECU) | Type of Anomaly |
| 1 | 288 | 12 | 3.5 | Positive | 5 | 295 | 12 | 3.1 | Positive |
| 2 | 289 | 12 | 0.5 | Positive | 6 | 296 | 12 | 7.9 | Positive |
| 3 | 293 | 12 | 3.9 | Positive | 7 | 297 | 12 | 4.7 | Positive |
| 4 | 294 | 12 | 8.6 | Positive | 8 | 298 | 8 | 12.6 | Positive |

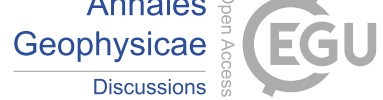

**Table 3.** MALZ Station Global Ionosphere Model Anomaly Table


| IONOLAB-TEC Anomaly Table for MALZ Station | | | | | | | | | |
|---|---|---|---|---|---|---|---|---|---|
| Number | DOY | Hour | TEC Difference (TECU) | Type of Anomaly | Number | DOY | Hour | TEC Difference (TECU) | Type of Anomaly |
| 1 | 288 | 12 | 2.3 | Positive | 5 | 296 | 12 | 2.5 | Positive |
| 2 | 293 | 12 | 0.4 | Positive | 6 | 298 | 6 | 8.6 | Positive |
| 3 | 294 | 10 | 7.4 | Positive | 7 | 304 | 0 | 0.2 | Negative |
| 4 | 295 | 10 | 3.6 | Positive | | | | | |

**Table 4.** MALZ Station IONOLAB-TEC Anomaly Table
Tables 3 and 4 show the anomalies found as a result of the analysis of the TEC values obtained
by the IONOLAB-TEC and GIM-TEC methods for the MALZ station. It is believed that the
positive anomaly on days 288 and 289 was caused by moderate (136.9 sfu, 150 sfu) solar
activity. It is also believed that the anomalies on the days 293, 294, 295 and 296 were caused
by strong (157.8 sfu, 166.3 sfu, 162.5 sfu, 153.9 sfu) solar activity.

Below are the TEC values for the OZAL station obtained using the GIM-TEC and IONOLAB-
TEC methods for the dates 13 October – 02 November (Figures 10 and 11).

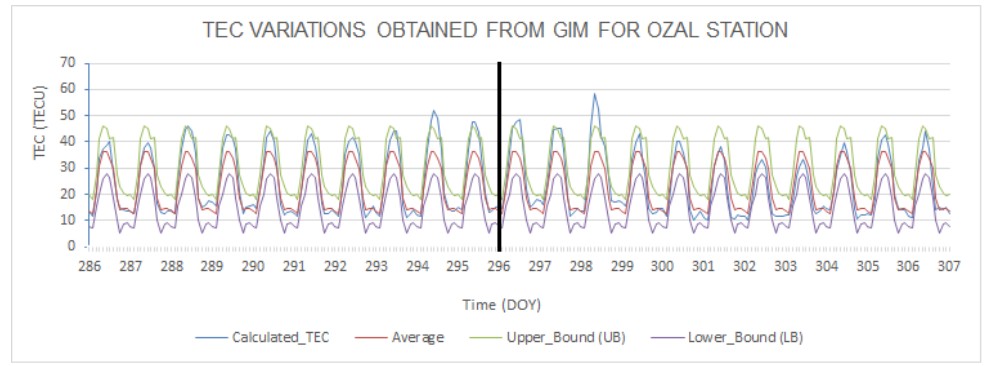


**Figure 10.** GIM-TEC Values for the OZAL Station





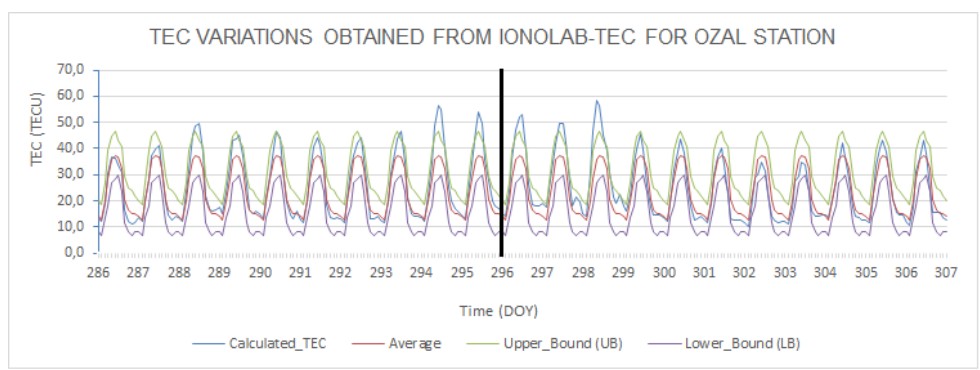


**Figure 11.** IONOLAB-TEC Values for the OZAL Station


The correlation coefficient *r* between the TEC values calculated by both methods for the OZAL
station was 0.982774 demonstrating a strong positive relationship. The anomaly tables for this
station are provided below (Tables 5 and 6).

| | | | | | | | | | | |
|---|---|---|---|---|---|---|---|---|---|---|
| **GIM-TEC Anomaly Table for OZAL Station** | | | | | | | | | | |
| **Number** | **DOY** | **Hour** | **TEC Difference (TECU)** | **Type of Anomaly** | | **Number** | **DOY** | **Hour** | **TEC Difference (TECU)** | **Type of Anomaly** |
| 1 | 288 | 12 | 2.8 | Positive | | 5 | 296 | 12 | 7.2 | Positive |
| 2 | 293 | 12 | 3.2 | Positive | | 6 | 297 | 12 | 4.0 | Positive |
| 3 | 294 | 12 | 7.9 | Positive | | 7 | 298 | 8 | 12.4 | Positive |
| 4 | 295 | 12 | 2.4 | Positive | | | | | | |

**Table 5.** OZAL Station Global Ionosphere Model Anomaly Table


| | | | | | | | | | | |
|---|---|---|---|---|---|---|---|---|---|---|
| **IONOLAB-TEC Anomaly Table for OZAL Station** | | | | | | | | | | |
| **Number** | **DOY** | **Hour** | **TEC Difference (TECU)** | **Type of Anomaly** | | **Number** | **DOY** | **Hour** | **TEC Difference (TECU)** | **Type of Anomaly** |
| 1 | 288 | 12 | 6.1 | Positive | | 7 | 295 | 10 | 7.4 | Positive |
| 2 | 289 | 12 | 1.6 | Positive | | 8 | 296 | 12 | 9.6 | Positive |
| 3 | 290 | 12 | 0.9 | Positive | | 9 | 297 | 12 | 6.0 | Positive |
| 4 | 293 | 12 | 3.5 | Positive | | 10 | 298 | 8 | 13.6 | Positive |
| 5 | 292 | 12 | 0.6 | Positive | | 11 | 301 | 14 | 1.2 | Negative |
| 6 | 294 | 12 | 11.8 | Positive | | 12 | 302 | 14 | 1.4 | Negative |

**Table 6.** OZAL Station IONOLAB-TEC Anomaly Table

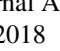


Below are the TEC values for the TVAN station obtained using the GIM-TEC and IONOLAB-
TEC methods (Figures 12, 13).

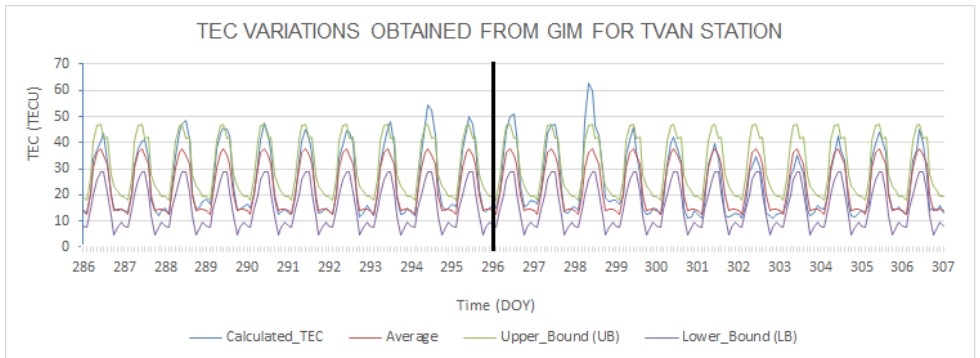

**Figure 12.** GIM-TEC Values for the TVAN Station

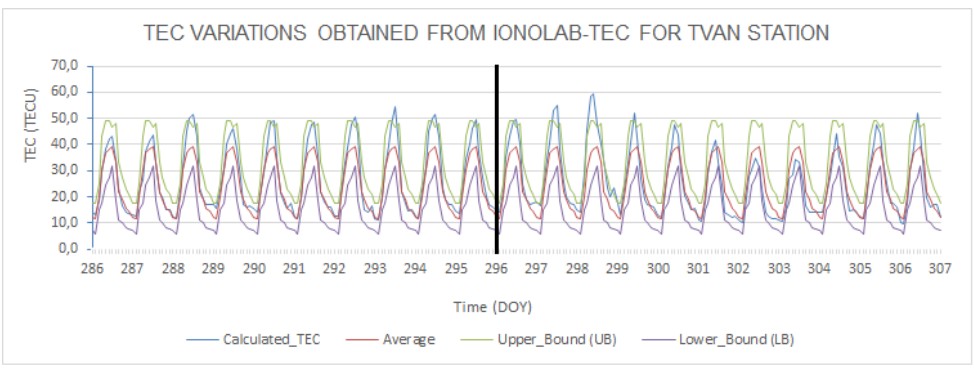

**Figure 13.** IONOLAB-TEC Values for the TVAN Station

The correlation coefficient between the TEC values calculated by both methods for the TVAN
station was 0.978363 representing a strong positive relationship. The anomaly tables for this
station are provided below (Tables 7 and 8).

| GIM-TEC Anomaly Table for TVAN Station | | | | | | | | | |
|---|---|---|---|---|---|---|---|---|---|
| Number | DOY | Hour | TEC Difference (TECU) | Type of Anomaly | Number | DOY | Hour | TEC Difference (TECU) | Type of Anomaly |
| 1 | 286 | 12 | 2.1 | Positive | 10 | 294 | 12 | 11.0 | Positive |
| 2 | 288 | 12 | 7.0 | Positive | 11 | 295 | 12 | 5.4 | Positive |
| 3 | 289 | 12 | 3.5 | Positive | 12 | 296 | 12 | 9.3 | Positive |
| 4 | 290 | 12 | 1.8 | Positive | 13 | 297 | 12 | 5.5 | Positive |
| 5 | 292 | 12 | 2.8 | Positive | 14 | 298 | 8 | 16.5 | Negative |
| 6 | 293 | 12 | 6.4 | Positive | | | | | |





**Table 7.** TVAN Station Global Ionosphere Model Anomaly Table


| IONOLAB-TEC Anomaly Table for TVAN Station | | | | | | | | | |
|---|---|---|---|---|---|---|---|---|---|
| Number | DOY | Hour | TEC Difference (TECU) | Type of Anomaly | Number | DOY | Hour | TEC Difference (TECU) | Type of Anomaly |
| 1 | 288 | 12 | 5.1 | Positive | 10 | 296 | 12 | 3.4 | Positive |
| 2 | 290 | 12 | 2.6 | Positive | 11 | 297 | 12 | 8.5 | Positive |
| 3 | 291 | 12 | 2.0 | Positive | 12 | 298 | 10 | 10.5 | Positive |
| 4 | 292 | 12 | 4.0 | Positive | 13 | 299 | 10 | 2.8 | Positive |
| 5 | 293 | 12 | 8.1 | Positive | 14 | 302 | 12 | 0.7 | Negative |
| 6 | 294 | 12 | 5.1 | Positive | 15 | 306 | 10 | 2.9 | Positive |
| 7 | 295 | 12 | 3.2 | Positive | | | | | |

**Table 8.** TVAN Station IONOLAB-TEC Anomaly Table



Tables 1, 2, 3, 4, 5, 6, 7 and 8 show the results of the statistical analysis of the TEC values
created by the IONOLAB-TEC and GIM-TEC methods. The tables also depict the day and hour
in which anomalies were observed, and the quantity and type of the anomaly. The numbers of
anomalies obtained in both models were very close to each other. The F10.7 cm index values
between the days 286 and 292 were 136 sfu, 135.4 sfu, 136.9 sfu, 150 sfu, 151.6 sfu, 145.7 sfu,
146.1 sfu. The index values show that there was usually moderate solar activity. Therefore, the
anomalies in question may be related to the earthquake or solar activity. The index values for
the days 293, 294, 295 and 296 (the day of the earthquake) were 157.8 sfu, 166.3 sfu, 162.5 sfu
and 153.9 sfu respectively. These values indicate strong solar activity. On the other hand, the
ionosphere layer was highly calm in these days in terms of geomagnetic conditions. As there
was strong solar activity, the numbers of anomalies were higher than the numbers in the days
286-292. Since solar activity was moderate in the day 297, the number of anomalies dropped.
The solar activity on the day 298 was moderate, but there was strong geomagnetic activity (Dst
-147 nt, Kp*10=73). The reason for the high numbers of anomalies on day 298 in both models
is believed to be due to geomagnetic activity. Considering the analyzed days in general, it may
be seen that it is difficult to identify earthquake-related anomalies as the solar activity and
geomagnetic conditions before and after the earthquake were not calm. Therefore, it is believed
that the anomalies detected in the stations on days 293-296 may be related to the earthquake
and/or solar activity, and the anomalies on days 297 and 298 may be related to the earthquake,
solar activity and/or geomagnetic activity.



## 4. CONCLUSION


In the scope of this study, the TEC values for the stations HAKK, MALZ, OZAL, TVAN were
obtained using the GIM-TEC and IONOLAB-TEC methods. In the comparison of the obtained
values, it was seen that there was high correlation between the TEC values obtained by the two
models. In order to detect earthquake-related TEC changes better, the TEC values created from
both models for the period of 13.10.2011-02.11.2011 were used as reference to determine the
UB and LB values. As a result of the statistical test, anomalies were found in all analyzed
stations for before, on the day of and after the earthquake. In order to understand whether the
anomalies obtained in both models were earthquake-related, the ionospheric conditions,
geomagnetic activity and solar activity on the analyzed days were examined using the Kp, Dst
and F10.7 cm indices.
Consequently, it was determined that the positive anomalies observed on days 286-292 may be
related to moderate solar activity and/or the earthquake, and the positive anomalies observed
on days 293, 294, 295, 296 (day of the earthquake) may be related to strong solar activity and/or
the earthquake. Moderate solar activity and strong geomagnetic activity were observed for day
298, so the numbers of anomalies in both models increased dramatically. This increase is
considered to be related to geomagnetic activity. The anomaly on day 298 may be related to the
earthquake, geomagnetic effects and/or solar activity. The finding that the ionospheric
conditions were vibrant in the analyzed days makes it highly difficult to identify earthquake-
related ionospheric changes. Therefore, interdisciplinary studies are needed to determine the
earthquake-related part of the change in question.

















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
