# Peer review of "MONITORING POTENTIAL IONOSPHERE CHANGES CAUSED BY"

_Annales Geophysicae, 2018_

## Referee Comment (RC1) · Anonymous Referee #1 · 20 Mar 2018

The idea to use simultaneously TEC derived by two different methods is interesting. However, authors are not familiar with the current state-of-the-art, the selected event is not good example for studying seismo-ionospheric effects due to simultaneous presence of solar and geomagnetic activity, the paper does not bring new knowledge in this area, and it suffers with various weaknesses (see below). Therefore I have to recommend reject the paper in its present form.

Major comments:

For your simple approach the Van earthquake is not suitable event for inferring effects of seismic activity, as it happened under not quiet solar and geomagnetic conditions.

[Figure]

In such situation much more sophisticated analysis has to be applied (e.g. Le et al., 2012). For this reason output of your analysis is inconclusive and it does bring any new knowledge.

When you study hourly values of TEC, you need to use a solar proxy with hourly time step, not F10.7 which provides one value for the day – e.g. satellite observations of EUV or X-rays.

Lines 331-334: The observed effects are rather standard effects of solar and geomagnetic activity.

Section Introduction repeats many well-known facts, which should be removed including Figs. 1 and 2. On the other hand, more references on seismo-ionospheric studies should be present including review paper by Pulinets and Davidenko (2014) and broad statistical study by Liu et al. (2010) and He and Heki (2017). I also recommend include in Introduction the paper by Akhoonzadeh et al. (2018) on multi-precursor analysis. Another interesting paper is that by Kelley et al. (2017).

Section 2.1 IONOLAB-TEC method could also be substantially reduced as all details are available in a well-accessible reference Arikan et al. (2004) in Radio Science.

Minor comments: - Line 56: "collision frequency of the ionosphere" should be "plasma frequency" as on line 58. - Line 64: Direct in situ measurements of electron density in the ionosphere are realized by satellites (e.g. Li and Parrot). - Lines 89-93. Another method of seismo-ionospheric investigations are VLF-LF measurements (e.g. Rozhnoi et al., 2015). - Lines 187-194. At present the official GIM-TEC maps of the International Global Navigation Satellite System Service (IGS) are created as an average from data/maps submitted by JPL, CODE, UPC and ESA, but maps produced by individual centers like CODE may also be used. - You use 1-10 October as calm days but the first days of October are not calm; e.g. 1 October is day of minor-to-moderate geomagnetic storm according to Fig. 4. - Figure 4: What does the scale for Kp mean? Sum Kp or scale should be 2o, 4o etc.? Individual 3-hour Kp values are defined within 1-9. -

Line 324: "ionosphere layer was highly calm" should be "ionosphere was calm"; "highly calm" means Kp less than1.

References: (if you do not have approach to below journals, use doi index or web address to get approach to abstract with e-mail address, and use the latter to ask author for a copy of the paper)

M. Akhoonzadeh et al.: Multi-precursor analysis associated with the powerful Ecuador (Mw= 7.8) earthquake of 16 April 2016 using Swarm satellites data in conjunction with other multi-platform satellite and ground data. Adv. Space Res., 61 (1), http://dx.doi.org/10.1016/j.asr.2016.12.004, 2018.

L. He, K. Heki: Ionospheric anomalies immediately before Mw = 7.0-8.0 earthquakes. J. Geophys. Res. Space Phys., 122 (8), 8659-8678, doi: 10.1002/2017JA024012, 2017.

M.C. Kelley et al.: Apparent ionospheric total electron content variations prior to major earthquakes due to electric fields created by tectonic stress. J. Geophys. Res. Space Phys., 122 (8), 6689-6695, doi: 10.1002/2017JA023601, 2017.

H.-M. Le et al.: A nonlinear background removal method for seismo-ionospheric anomaly analysis under a complex solar activity scenario: A case study of the M9.0 Tohoku earthquake. Adv. Space. Res., 50 (2), 211-220, doi: 10.1016?j.ar.2012.04.001, 2012.

M. Li, M. Parrot: Statistical analysis of the ion density recorded by DEMETER in the epicewnter areas of earthquakes as well as in their magnetically conjugated areas. Adv. Space Res., 61 (3), 974-984, https://doi.org/10.1016/j.asr.2017.10.047, 2018.

J.-Y. Liu et al.: A statistical study of ionospheric earthquake precursors monitored by using equatorial ionization anomaly of GPS TEC in Taiwan during 2001-2007. J. Asian Earth Sci., 39, 76-80, doi: 10.1016/j.seases.2010.02.012, 2010.

S. Pulinets, D. Davydenko: Ionospheric precursors of earthquakes and Global electric

circuit. Adv. Space Res., 53 (5), 709-723, doi: 10.1016/j.asr.2013.12.035, 2014.

A. Rozhnoi et al.: VLF/LF signal studies of the ionospheric response to strong seismic activity in the Far Eastern region combining the DEMETER and ground-based observations. Phys. Chem. Earth, 85-86, 141-149, http://dx.doi.org/10.1016/j.pce.2015.02.005, 2015.

---

## Author Comment (AC1) · 20 Mar 2018

Line 56: it was edited as plasma frequency Line 58-64: Äřt was referenced as you cite Line 89-93: Äřt was referenced as you cite Line 187- 194: In this study we used GIM published by CODE FÄřG4 : It says (Kp*10) and it was edited Fig 1 and Fig 2 were removed from the paper as you suggest. You suggest some refereences, we have added it to our paper. You have also mentioned about that Van earthquake is not suitable to detect ionospheric anomaly induced by earthquake. As you know, There is not much work on this earthquake. Therefore we would like to study this earthquake. In addition to that we want to see ionospheric variations with two different method. As far as I know, there is no study using two different method in this study. We think that this paper presents this originality.

Please also note the supplement to this comment:
https://www.ann-geophys-discuss.net/angeo-2018-11/angeo-2018-11-AC1-supplement.pdf
* * *
[Figure]

**Supplement:**

| 3 | MONITORING OF POSSIBLE IONOSPHERIC DISTURBANCES |
|---|-------------------------------------------------|
| 4 | CAUSED BY VAN EARTHQUAKE (Mw 7.2) USING GNSS    |
| 5 | MEASUREMENTS                                    |

**6 7 8 Selcuk PEKER1, Samed INYURT 2and Cetin MEKIK2**

General Command of Mapping, Ankara, Turkey (selcuk-peker@hotmail.com)

2Bulent Ecevit University, Geomatics Engineering Department, Zonguldak (samed\_inyurt@hotmail.com, cmekİk@hotmail.com)

**11 **ABSTRACT**

[revised manuscript text omitted]

$$H_k(\omega_c) = \begin{cases} 1, & \text{if } \pi - \omega_c \le \frac{2\pi}{N} k \le \pi + \omega_c \\ 0, & \text{diger} \end{cases}$$
(10)

$$h_n(k_c) = \begin{cases} 1 - \frac{1}{N}(2k_c + 1), & n = 0\\ -\sin\left(\frac{\pi n}{N}(2k_c + 1)\right) / \left(N\sin(\frac{\pi n}{N})\right), & n \neq 0 \end{cases}$$
(11)

The error function between the VTEC values calculated from satellites  $\mathbf{x}_m$  and the VTEC 135 estimates  $\widetilde{\mathbf{x}}$  is given in Equation (12). The operation  $\|.\|$  describes the norm statement of the 136 difference vector weighted between the VTEC estimates and calculations.

$$e(\mu, k_c) = \sum_{\mathbf{m}=1}^{M} \|\mathbf{W}_{\mathbf{m}}(\mathbf{\tilde{x}} - \mathbf{x}_{\mathbf{m}})\|^2$$
(12)

In order to regularize the estimate values even more, floating median filter may be used. The length of the median filter is another parameter to be determined. With the estimated VTEC values, post-estimation median filter was applied, and the error function between the VTEC values is given in Equation (13).

$$e_f(N_f) = \left\| \mathbf{\tilde{x}} - \mathbf{\tilde{x}}_{\mathbf{N}_f} \right\|^2$$
(13)

In Equation (13),  $\tilde{\mathbf{x}}_{N_f}$  shows the  $\tilde{\mathbf{x}}$  estimates processed with a median filter with the length of *Nf*. For the method to work accurately, suitable  $\mu$ ,  $k_c$  and  $N_f$  parameters must be determined. The details provided up to now cover the regularization method for a period of 24 hours.

When there is an estimation of TEC for a limited period of time, the cost function is redefinedas in Equation (14).

$$J_{\mu,k_c}(x) = \sum_{m=1}^{M} (x - x_m)^T W_m(x - x_m) + \mu(x - at)^T H(k_c)(x - at)$$
(14)

In the equation, a is the slope of the line and **t** is the time vector for the period of time. In order to find **x** estimates that minimize the cost function, the derivative of this function is taken, and the result is equated to zero. In this case, minimization of the cost function is turned into the solution of a system of equations as in Equation (15).

$$A(\boldsymbol{\mu}, \boldsymbol{k}_{c}) \begin{bmatrix} \boldsymbol{x} \\ \boldsymbol{a} \end{bmatrix} = b$$
 (15)

- 160 The matrix **A** in Equation (16) and the vector **b** in Equation (17) are calculated as,
- 161

$\mathbf{A}(\mu,k_c) = \begin{bmatrix} \sum_{m=1}^{M} \mathbf{W}_m + \mu \mathbf{H}(k_c) & -\mu \mathbf{H}(k_c) \\ \mathbf{t}^T \mathbf{H}(k_c) & -\mathbf{t}^T \mathbf{H}(k_c) \mathbf{t} \end{bmatrix}$ (16)

$$\mathbf{b} = \begin{bmatrix} \sum_{m=1}^{M} & \mathbf{W}_m \mathbf{x}_m \\ & 0 \end{bmatrix}$$
(17)

Using the equations above, the  $\tilde{\mathbf{x}}$  values showing the  $\mathbf{x}$  estimates are calculated as in Equation 167 (18).

$$\begin{bmatrix} \tilde{\mathbf{x}}(\boldsymbol{\mu}, \boldsymbol{k}_c) \\ a \end{bmatrix} = A^{-1}(\boldsymbol{\mu}, \boldsymbol{k}_c) \boldsymbol{b}$$
(18)

As a result, the proposed regularization method may be applied for both day-long and limitedperiods of time (Arıkan et al. 2004).

**174 **2.2 Global Ionosphere Model (GIM):**

Global Ionospheric Maps are published in the IONEX (IONosphere map EXchange) format in 177 a way that covers the entire world. The institutions that produce these maps in the world include 178 CODE (Center for Orbit Determination in Europe, Switzerland), DLR (Fernerkundungstation 179 Neustrelitz, Germany), ESOC (European Space Operations Centre, Germany), JPL (Jet 180 Laboratory, California), NOAA (National Oceanic and Atmospheric Propulsion 181 Administration, United States), NRCan (National Resources, Canada), ROB (Royal 182 Observatory of Belgium, Belgium), UNB (University of New Brunswick, Canada), UPC 183 (Polytechnic University of Catalonia, Spain), WUT (Warsaw University of Technology, 184 Poland) (Aysezen, 2008). In this study we used the GIM-TEC values produced by CODE in the 185 IONEX format. In the dates they were analyzed, the temporal resolution of the TEC values was 186 2 hours, while their positional resolution was 2.5° by latitude and 5° by longitude. In order to 187 calculate TEC values for a point whose latitude and longitude is known on the GIM-TEC maps 188 created by CODE using more than 300 GNSS receivers around the world, the 4 TEC values

- 189 that cover the point and the two-variable interpolation formula are given below (Schaer et al.
- 190 1998).

$$E_{int}(\lambda_0 + p\Delta\lambda, \beta_0 + q\Delta\beta) = (1-p)(1-q)E_{0.0} + p(1-q)E_{1.0} + q(1-p)E_{0.1} + pqE_{1.1}$$
 (19)

p and q:  $0 \le p, q

205

**Figure 1. Analyzed Stations**

Figure 3 shows the stations analyzed (represented by red triangles) and the epicenter of the earthquake represented by blue star. For each station, the TEC values with the temporal resolution of two hours obtained from both the IONOLAB-TEC and GIM-TEC models and the correlation coefficient showing whether there is a linear relationship between two values were calculated as below;

$$r = \frac{\sum (xy) - (\sum x)(\sum y)/n}{\sqrt{(\sum x^2 - (\sum x)^2/n)(\sum y^2 - (\sum y)^2/n)}}$$
(20)

In order to determine the outlier values among the TEC values with a two-hour temporal 215 resolution from both models, the TEC values obtained from both models between the dates 216 01.10.2011 and 10.10.2011, which were considered calm in terms of geomagnetic and solar 217 activity, were used to determine the upper boundary (UB) and the lower boundary (LB). By 218 utilizing the TEC values from both models, the UB and LB values were calculated using the 219 formulae x+3 $\sigma$  and x-3 $\sigma$ . Here, x is the mean TEC value for the relevant epoch and  $\sigma$  is the 220 standard deviation. If the TEC value in any epoch is higher than the upper boundary, it is a 221 positive anomaly. Similarly if it is lower than the lower boundary, it is a negative anomaly. In 222 order to investigate whether the anomalies before, on the day of and after the earthquake were 223 caused by the earthquake or not, we also examined the (Kp\*10), Dst and F10.7 cm indices, 224 which provided information on the geomagnetic and solar activity for the days in which 225 anomalies were detected.

---

## Referee Comment (RC2) · Anonymous Referee #2 · 26 Mar 2018

General Review:

This manuscript presents ionospheric total electron content (TEC) derived from two different methods (Ionolab-TEC and CODE GIM-TEC) for GPS stations near the Van earthquake (Mw 7.2) in Turkey. Unfortunately, I cannot recommend publication of this manuscript in Ann. Geophys. because the paper fails to present any new results of TEC changes related to earthquakes.

Major Comments:

The authors make no attempt to sort out possible geomagnetic, solar, and earthquake effects. Their conclusion is very weak: "As the ionospheric conditions in the analyzed

days were highly vibrant, it was thought that the anomalies were caused by geomagnetic effects, solar activity and the earthquake." Moreover, the manuscript presents a large amount of data with little analysis.

In order for publication, the authors would have to show that the Ionolab-TEC method is better at identifying possible earthquake related signals than other methods. This paper just shows that their method is about the same as the CODE GIM-TEC method. And both methods fail to identify any interesting signals possibly related to the earthquake. I don't see any results that justify publication.

The authors also need to reference and discuss studies that both support and fail to support TEC earthquake precursory signals. The field is extremely controversial. For example,

Heki, K. and Enomoto, Y.: Mw dependence of the preseismic ionospheric electron enhancements, J. Geophys. Res.-Space, 120, 7006–7020, https://doi.org/10.1002/2015JA021353, 2015.

Kamogawa, M. and Kakinami, Y.: Is an ionospheric electron enhancement preceding the 2011 Tohoku-Oki earthquake a precursor?, J. Geophys. Res.-Space, 118, 1751–1754, https://doi.org/10.1002/jgra.50118, 2013.

Thomas, J. N., Huard, J., and Masci, F.: A statistical study of global ionospheric map total electron content changes prior to occurrences of $M \geq 6.0$ earthquakes during 2000–2014, J. Geophys. Res.-Space, 122, 2151–2161, https://doi.org/10.1002/2016JA023652, 2017.

Masci, F., Thomas, J. N., Villani, F., Secan, J. A., and Rivera, N.: On the onset of ionospheric precursors 40 min before strong earthquakes, J. Geophys. Res.-Space, 120, 1383–1393, https://doi.org/10.1002/2014JA020822, 2015.

Minor Comments:

Figures 1 and 2 are not needed. Referencing the papers these figures came from is

sufficient.

The Ionolab-TEC method is described in fine detail. This is not needed. Referencing the papers that describe this method is sufficient.

More concise figures are needed to summarize their results. Showing all the data is good, but can be in an appendix or supplemental materials section.

---

## Author Comment (AC2) · 26 Mar 2018

Fig 1 and Fig 2 were removed from the paper as you suggest. You suggest some refereences, we have added it to our paper. You have also mentioned about There is no attepmt to solve solar, magnetic and earthquake effect. We would like to analyze as statistical and we think that we are succesfull on this stage. We dont want to show anomalies like only caused by earthquake like other wrong papers. Ärt will be not convenient for us. The other thing is that we dont need to show Ionolab method is better than GIM-TEC models. Ärt is not our subject. We only discuss that there is an anomaly or not before during and after the earthquake using two different models. We

also believed that we explained detaily results in conclusion